# Extraction and Characterization of Artichoke (*Cynara cardunculus* L.) Solid Waste from the Industrial Processing of Fresh-Cut Products for Nutraceutical Use

**DOI:** 10.3390/foods14010013

**Published:** 2024-12-25

**Authors:** Francesco Corrias, Efisio Scano, Massimo Milia, Alessandro Atzei, Mattia Casula, Nicola Arru, Alberto Angioni

**Affiliations:** 1Food Toxicology Unit, Department of Life and Environmental Science, University of Cagliari, University Campus of Monserrato, 09042 Cagliari, Italy; max_milia@hotmail.it (M.M.); alessandro.atzei@unica.it (A.A.); mattiacasula92@gmail.com (M.C.); nicola.arru.logica@gmail.com (N.A.); 2Faculty of Agraria, University of Sassari, 07100 Sassari, Italy; efisiscano@gmail.com

**Keywords:** artichoke, polyphenols, inulin, antioxidant

## Abstract

Artichoke (*Cynara cardunculus* L.) is an herbaceous perennial plant from the Mediterranean Basin, cultivated as a poly-annual crop in different countries. Artichoke produces a considerable amount of waste at the end of the harvesting season in the field (5.2 tons/ha/year, DW) and from the industrial processing of fresh-cut products during the harvesting time (800 tons/year). The qualitative and quantitative phenolic profile and inulin content of artichoke samples from the field and industrial processing waste have been investigated after green extraction. The best operative conditions were achieved using the dried biomass extracted with water at 80 °C for 120 min and a matrix-to-solvent ratio of 1:30. The data obtained showed that the concentration of total polyphenols in fresh artichokes followed this order: stems > heads > leaves > outer bracts. Chlorogenic acid and 3,4 di-O-caffeoylquinic acid were the most concentrated caffeoylquinic derivates, whereas luteolin 7-O-malonyglucoside, luteolin 7-O-glucoside and 7-O-rutinoside were the most abundant flavonoids. The artichoke by-products showed high polyphenolic and inulin values, thus representing an important source of health-promoting biomolecules for application in pharmaceutical and cosmeceutical fields. According to the principles of circular economy, the work scheme proposed in this article, the use of waste and its processing into useful products such as nutraceutical supplies could provide a practical and economic opportunity for companies and farmers involved in the cultivation and industrial processing of artichokes.

## 1. Introduction

Artichokes (*Cynara cardunculus* L. var *scolymus*) are highly appreciated vegetables belonging to the Asteraceae family. Artichokes are native to the Mediterranean Basin and are cultivated worldwide due to their ability to grow in different soils and climate conditions, with average yields ranging from 5 to 30 tons/ha/year, depending on the cultivar and the environmental conditions [1,2]. The total worldwide production of artichokes averages 1700 Ktons/year, with Italy producing about 438 Ktons in 2022 and representing the primary producer worldwide (39% of total world production) [2,3]. Sicily, Puglia, and Sardinia account for 90% of the total cultivated Italian area and 85% of the national production [1]. Globe artichoke plants are characterized by green, deeply lobed leaves and flower heads consisting of green/violet bracts surrounding the receptacle, known as the heart. Artichokes are consumed raw, cooked, or canned; the heart, the bracts surrounding it (capitulum), and the soft part of the stem represent their edible part. Conversely, the hard part of the stem, the outer bracts, the roots, the stalks, and the leaves represent the non-edible parts (waste) [4,5]. Fresh-cut products are defined as fresh fruits and vegetables subjected to technological processes of minimal entity, packaged and readied for human consumption soon after harvest [6].

The industrial processing of artichokes for fresh-cut products contemplates only the use of the receptacle and the capitulum, with a considerable amount of waste generated yearly from the field at the end of the season (5.2 tons per hectare DW) and from the industrial processing of fresh-cut products during the harvesting time (800 tons per year). Two types of waste products are generated: one is composed of the non-edible bracts and stems (15–30%), and the other by the remaining part of the plant (leaves, stalks, and roots) left on the field (70–85%). About 10% of the non-edible part is used as feed for sheep and cows; however, the prevalent part is waste [7,8,9,10,11].

Artichokes are considered functional foods with health-promoting, pharmacological, and nutraceutical characteristics. They have a high concentration of bioactive compounds, which accumulate in the outer parts during plant development; these include polyphenols, inulin, vitamins, minerals, and fiber. Many authors have investigated the distribution of polyphenols in the plant parts, highlighting significant differences. Caffeoylquinic acids and flavones are the most represented polyphenols and are involved in defense against abiotic stress and pathogens [12,13,14,15,16,17,18,19,20]. In addition, these compounds have antioxidant activity as reactive oxygen species (ROS) inhibitors and contribute to health-promoting, antilipidemic, diuretic, anti-inflammatory, and hepatoprotective effects, decreasing the incidence of cardiovascular diseases [21,22,23,24,25,26,27,28,29,30]. Inulin is a natural, plant-derived compound that aids in the storage of nondigestible oligosaccharides composed of the oligo- and polysaccharides of fructose units with a low caloric value. Inulin acts as dietary fiber, improving the well-being of the gastrointestinal tract; it is more concentrated in the roots, with concentrations ranging from 5 to 45% DW [15,31,32,33,34,35,36,37]. The concentration of polyphenols and inulin is strongly influenced by different endogenous and exogenous factors, such as the cultivar, age, pre-harvest and post-harvest processes, and environmental conditions [38,39,40,41,42].

Some authors have reported the influence of industrial processes on waste, not considering the products deriving from the field [16]; others have conducted studies on plants produced in the field and/or bought on the market in order to avoid industrial waste [8,14]. For the first time, the present study characterized the content of bioproducts (polyphenol compounds and inulin) in the various portions of the artichoke cultivar Tema (outer and inner bracts, stem, choke, heart, and roots), which were collected from the industrial supply chain of fresh-cut artichoke products, and from the waste collected during the processing step and left in the field after harvest.

Green technologies and smart solvents are acquiring importance due to environmental concerns, by reducing the chemicals used in conventional extraction techniques. Further, data in the literature have reported their use for the extraction of bioactive compounds from plants and plant-based material, reducing the extraction time and increasing yields [43,44].

Fresh and dried samples were subjected to green extractions, and the feasibility of recovering bioactive compounds for use in the pharmaceutical, food, and cosmetic industries was evaluated.

## 2. Materials and Methods

### 2.1. Chemicals and Reagents

Methanol, acetonitrile, orthophosphoric acid (85%), hydrochloric acid (37%), sulfuric acid (96%), glucose, fructose, phenol, sodium carbonate, and Folin–Ciocalteu reagent were purchased from Carlo Erba (Milan, Italy). In addition, 3,5-dinitro salicylic acid was purchased from Thermo Scientific (Rodano, Milan, Italy). Inulin, apigenin, caffeic acid, luteo-lin-7-O-glucoside, luteolin-7-O-glucuronide, luteolin-7-O-rutinoside, luteolin, 1-O-caffeoylquinic acid, 3-O-caffeoylquinic acid, 5-O-caffeoylquinic acid (chlorogenic acid), 1,3-di-O-caffeoylquinic (cynarine) and 3,4-di-O-caffeoylquinic acid were all from Sigma-Aldrich (Milan, Italy).

Double-deionized water was produced with a Milli-Q system, showing a conductivity of less than 18.2 MΩ (Millipore, Bedford, MA, USA).

### 2.2. Plant Material

Artichokes of the cultivar Tema were cultivated in industrial fields in Samassi (Sardinia, Italy) with a 1 × 1 m^2^ plant space. Two harvests were carried out: the first was during the period of maximum production and the second was at the end of the harvesting season, using the plants left in the field. Fifty fresh artichokes were randomly selected in each harvest and taken to the laboratory for processing, then divided into three subsamples for replicate analyses. The remaining samples harvested from the field were brought to the processing plant for industrial processing.

The fresh artichokes brought to the laboratory were washed with tap water and divided into heads (receptacle + capitulum) (F_H_), outer bracts (F_B_), stems (F_S_), and leaves (F_L_). Each portion was finely cut by hand and then chopped and homogenized using a 600 W stainless steel food blender (Girmi, Rimini, Italy).

The waste from the industrial processing (~18 Kg) was mixed carefully and divided into nine homogeneous sub-samples (~2 Kg/each); three sub-samples (A) were dried at 30 °C in an oven for three days without grinding (whole dry waste: WD_W_), whereas the remaining six sub-samples (B) were ground using a bio-shredder apparatus (Viking GE 105, Stihl, Italy). Samples B were analyzed a) fresh (fresh waste: F_W_) and b) dried at 30 °C in an oven for three days (dry waste: D_W_).

The plants from the field were collected fresh (FP) and at the end of the harvesting season after being dried in the sun (whole dry plant: WD_P_). FP was divided into stem + leaves (FP_SL_) and roots (FP_R_).

Each portion (F_H_, F_B_, F_S_, F_L_, WD_W_, F_W_, D_W_, FP_SL_, FP_R_, WD_P_) was subjected to the analytical procedures for bioactive compound determination. Each analytical determination was carried out in triplicate.

### 2.3. Moisture and Ash Evaluation

Five grams of homogenized samples were dried at 105 °C until a constant weight was obtained (about 24 h). The samples were then carbonized at 550 °C in a porcelain crucible for 8 h for ash quantification. The moisture and ash contents were expressed as % fresh weight (%FW).

### 2.4. Polyphenol Extraction Procedure

The artichoke samples (50 g) were subjected to two extraction procedures for polyphenol analysis. Conventional extraction was performed with MeOH/H_2_O (80/20) at ambient temperature (rate 1/30) and green extraction was performed with MilliQ water at 80 °C under constant stirring in 250 mL glass flasks. Different matrix-to-solvent ratios (1:5, 1:10, 1:20, and 1:30) and extraction times (40 and 120 min) were tested on FB samples to evaluate the best hot water extraction approach. The extracting solvents were centrifuged at 3154× *g* and 10 °C for 15 min by using a Centrifuge 5810 R (Eppendorf, Hamburg, Germany), filtered with 0.45 μm PTFE syringe filters (Thermo Scientific, Rome, Italy) and transferred to a 1.8 mL vial for LC-DAD analysis.

### 2.5. Polyphenols HPLC—DAD Analysis

An Agilent HPLC 1100 system coupled with a diode array detector (DAD), controlled by an Agilent ChemStation (Agilent Technologies, Rome, Italy) was used for polyphenol characterization and quantification. The column was a Phenomenex C18 (5 μm, 150 mm × 4.6 mm) working at room temperature. A binary solvent gradient was employed: (A) H_3_PO_4_ 0.22 M, and (B) ACN/CH_3_OH (1/1, *v*/*v*). The following program was used: t = 0 96% A, t = 40 50% A, t = 50 40% A, post time 5 min. The injection volume and the flow rate were 20 μL and 0.4 mL/min, respectively. The caffeoylquinic acid and apigenin derivatives were monitored at 313 nm, whereas luteolin derivates were monitored at 360 nm.

Caffeoylquinic acids were expressed as chlorogenic acid equivalents, whereas apigenin’s and luteolin’s derivates were expressed as apigenin and luteolin equivalents, respectively. The polyphenol characterization compared the retention times and UV-VIS spectra with authentic standards and data available in the literature. The quantification data were reported as mg/kg FW.

### 2.6. Inulin and Carbohydrates Extraction and Analysis

First, 100 g of artichoke waste sample (D_W_, WD_W_, WD_P_, and FP_R_) was extracted with 3 L of water at 80 °C for 120 min. One liter of the extracts was centrifuged (3154× *g* and 10 °C for 20 min), filtered on a Büchner funnel (Whatman cellulose filter paper 10 μm pore size), and concentrated in rotavapor (65 °C) up to 50 mL, and subjected to two freezing and thawing cycles. After centrifugation at 3154× *g* and 5 °C for 45 min, the pellet was dried at 50 °C until a constant weight was obtained [32].

The amount of inulin was determined by calculating the absolute weight of the pellet with respect to the fresh weight and was compared to the inulin content calculated by the difference between the total carbohydrates and reducing sugars [33].

Total carbohydrates were determined by the phenol–sulfuric acid method [45]. Briefly, 20 mg of sample was placed in a 15 mL glass tube with 5 mL of HCl 1 M and heated for 2 h at 100 °C. In another 15 mL glass tube, 1 mL of the hydrolyzed extract was added to 1 mL of phenol at 5% in water (*w*/*v*) and 5 mL of H_2_SO_4_ (96%) and shaken gently for 1 min at room temperature. The solution was left to rest for 30 min in the dark. Finally, the sample was read at a wavelength of 488 nm against a control blank (H_2_O, phenol 5% and H_2_SO_4_) by using a spectrometer (Cary 50, Varian Inc., Agilent, Milan, Italy). Total carbohydrates were expressed as g/100 g (FW) of equivalent in D-glucose.

Reducing sugars were calculated according to the dinitro salicylic (DNS) acid method [46]. The DNS reagent was prepared by dissolving 1 g of DNS in 20 mL of NaOH 2 M (solution A) and 30 g of sodium/potassium tartrate tetrahydrate in 50 mL of distilled water (solution B). After that, solution A and B were mixed and diluted with distilled water until a volume of 100 mL was reached. Then, 1 mL of artichoke extract and 1 mL of DNS reagent were pooled in a 10 mL glass tube and put in a bain-marie at 100 °C for 5 min. The tube was left to cool at room temperature and then the volume of the tube was increased with 8 mL of distilled water. The sample was read at a wavelength of 540 nm by using a spectrometer. Reducing sugars were expressed as g/100 g (FW) of equivalent in fructose.

### 2.7. Statistical Analysis

Analysis of variance (ANOVA) was performed using the software XLSTAT (Addinsolf Ltd. (Paris, France), Version 19.4). Mean comparisons were analyzed by using multiple *t*-tests with Bonferroni–Dunn correction for multiple comparisons (*N* = 3).

## 3. Results and Discussion

### 3.1. Moisture and Ash

The fresh samples (F_H_, F_B_, F_S_, F_W_, FP_SL_) showed average moisture values of 83.4 ± 2.3 (% ± RSD%), ranging from 81.2% (FP_SL_) to 85% (F_S_) (Table 1). The leaves (F_L_) and the roots (FP_R_) showed values averaging 76.1 ± 0.3 and 62.2 ± 1.2 (% ± RSD%), respectively (Table 1). The FP_SL_ values overlapped the mean of F_S_ + F_L_. The dried samples, WD_W_, D_W_, and WD_P_, showed similar values averaging 15.1 ± 11.9 (% ± RSD%); among these, D_W_ showed the lower values (12.4%) (Table 1). The difference between WD_W_ and D_W_ can be attributed to the higher surface/weight ratio of the chopped vegetables, which allows for higher water evaporation.

Ashes ranged from 1.0 ± 9.8% to 1.4 ± 10.6% in all samples, but showed a higher content in F_L_ (1.8% ± 16.8%) and FP_R_ (1.6 ± 0.7%) (Table 1).

### 3.2. Polyphenols Characterization and Quantification

The HPLC/DAD analytical determination allowed the identification of fifteen polyphenols belonging to three chemical families (caffeoylquinic acid, luteolin, and apigenin) (Figure 1). The extraction of polyphenols is commonly carried out by liquid–liquid or liquid–solid methods using pure organic solvents or mixed with HPLC water [47]. Artichoke is particularly rich in fiber and requires effective extraction to disrupt the matrix and recover the analytes.

Green extraction with hot water (80 °C) was performed on fresh samples to evaluate the best matrix/solvent ratios and extraction times. Samples of homogenized F_B_ were extracted in hot water with a matrix-to-solvent ratio of 1:5 for 40 and 120 min. After 40 min of maceration, the total polyphenols value accounted for 4528.9 ± 4.0 (mg/kg ± RSD%), which increased to 5191.4 ± 3.5 after 120 min (Table 2). By increasing the matrix-to-solvent ratio to 1:10, 1:20, and 1:30 and maintaining the extraction time at 120 min, the values of the total polyphenols accounted for 5381.8 ± 3.8, 5773.5 ± 4.4, and 6767.6 ± 2.3 (mg/kg ± RSD%), respectively. The best results were obtained with a ratio of 1/30 and a time of 120 min (Table 2). Caffeoylquinic acids were more concentrated than flavonoids in all extraction ratios and times (on average, 85.8 ± 1.2% vs 13.6 ± 4.5%). Among the caffeoylquinic acid congeners, chlorogenic acid was the most represented (28.1 ± 2.6%), accounting for 1901.5 ± 4.9 mg/kg ± RSD%, followed by 3–4 di-O-caffeoylquinic acid 1536.8 ± 8.2 mg/kg ± RSD% (22.7 ± 8.8%), and cynarine 634.5 ± 4.8 mg/kg ± RSD% (9.4 ± 3.3%). Luteolin 7-O-rutinoside and luteolin 7-O-malonyglucoside were the most concentrated flavonoids, accounting on average for 3.4 ± 0.4%, followed by luteolin 7-O-glucoside 2.4% (Table 2).

The selected green water extraction was compared with the standard MeOH/H_2_O extraction on fresh samples of F_B_, F_S_, F_H_, and F_L_ (Table 3). In all samples, the extraction with hot water showed higher extraction rates for total polyphenols, with increases ranging from 171.1 ± 5.3% (F_H_ and F_s_) to 314.2% (F_L_) (Table 3). Therefore, all samples were extracted using hot water with the selected matrix/solvent ratio, time, and temperature (Table 3 and Table 4). Moreover, in the MeOH/H_2_O extracts, the 4-O-caffeoylquinic acid overlapped with chlorogenic acid and was not detected. The caffeoylquinic acids were the most represented, with overlapping values in F_S_, F_H_, and F_L_ (in average 12,806.4 ± 2.0), whereas F_B_ showed values almost half of the previous (6767.6 ± 2.3) (Table 3). Chlorogenic acid and 3–4 di-O-caffeoylquinic acid were the most concentrated caffeoylquinic acids.

The number of single phenols detected was different in the various parts of the artichoke. 1-O-caffeoylquinic acid was more abundant in F_H_, whereas 3,4 di-O-caffeoylquinic acid in F_H,_ F_B_ and F_L_ and 1,5-di-O-caffeoylquinic acid in F_S_ and F_H_ were more abundant (Table 3). Flavonoids showed overlapping values, luteolin was found in low quantities and was mainly conjugated, luteolin 7-O-rutinoside was the most concentrated in all samples, and luteolin 7-O-glucoside had higher values in the leaves (Table 3). Luteolin 7-O-glucuronide showed a low value and was not detected in the samples extracted with MeOH/H_2_O; a similar behavior was shown in apigenin.

The data obtained for the samples subjected to the drying process showed a lower apparent concentration of polyphenols compared to the fresh samples. WD_W_ and D_W_ accounted for 39.8 ± 8.0% and 19.4 ± 7.6% of the F_W_ (Table 4). However, considering the conversion factor of dry/fresh related to the water content in fresh samples, dried samples resulted in the greater extraction of polyphenols.

The grinding process led to a greater decrease in all compounds. This process has been found to reduce the antioxidant activity of polyphenols due to their degradation, as in the case of cereal bran, fruit peel, and tomatoes [47], probably due to the de-compartmentalization and contact of the phenolic substrates present in the vacuoles with the cytoplasmic oxidases [48,49]. Sun-dried samples (WD_P_) were processed whole (leaves, stem and roots), and the number of total polyphenols was between the values found in WD_W_ and D_W_. F_W_ is mainly made of stems, bracts, and leaves; by comparing the levels of flavonoids with the analysis of the fresh parts, it can be noted that the contribution of the stems and bracts is much more significant than that of the leaves in defining the amounts of total flavonoids in this fraction from the industrial process. FP_SL_ and F_W_ showed overlapping values for the total and single polyphenols, and a similar behavior was evinced also for WD_W_ and WD_P_ (Table 4). The roots (FP_R_) showed lower values for all polyphenol compounds, even if the cynarine amounts were in line with the fresh fractions. Concerning flavonoids, lower values were detected for luteolin 7-O-rutinoside and higher values were detected for luteolin 7-O-malonylglucoside.

### 3.3. Inulin and Carbohydrates Quantification in Artichoke By-Products

The samples collected for the analysis of inulin were the dried extracts produced in the industrial process, the whole sun-dried plant, and the fresh roots.

D_W_, WD_W_, WD_P_, and FP_R_ showed inulin values accounting for 2.7 ± 17.8 (g/100 g ± RSD%), 2.2 ± 5.5 (g/100 g ± RSD%), 1.7 ± 13.5 (g/100 g ± RSD%), and 5.1 ± 17.1 (g/100 g ± RSD%) by weight, and total carbohydrates showed values accounting for 3.7 ± 14.2, 3.5 ± 15.8, 3.1 ± 7.9 and 7.2 ± 6.4 (g/100 g ± RSD%), respectively (Figure 2).

Reducing sugar accounted for 1.2 ± 13.8, 0.9 ± 7.6, 1.1 ± 9.7, and 2.2 ± 7.0 (g/100 g ± RSD%), respectively. The values obtained by subtracting the total inulin from total carbohydrates overlapped with those of the reducing sugars. Therefore, the main part of the carbohydrate fraction of artichokes was composed of inulin (Figure 2).

## 4. Discussion

Artichoke is a well-known product that can be used as a functional food and source of nutraceutical ingredients [7]. It has a high moisture content, particularly in the stem, flower head, and leaves, with the roots being the driest portion. Our data were consistent with what was reported in the literature, with moisture values ranging from 82% to 85% in the fresh parts, except for the leaves and the roots that had lower moisture values [50,51]. According to previous studies, the qualitative characterization of the polyphenolic fraction of the different portions of the artichoke has shown a profile characterized by the presence of caffeoylquinic acid derivatives and flavonoids (luteolin derivatives, luteolin, and apigenin), accounting for 9–12 main biomolecules depending on the artichoke fraction [4,5,7,8,9,10,11,12,13,14].

The assessment of total polyphenols in fresh artichokes showed average increased values in the order stems > heads > leaves, with overlapping values after statistical evaluation. In contrast, the outer bracts showed the lowest levels (Table 3). Different authors reported similar values, with total polyphenols following the order: leaves > heads > floral stems > outer bracts [16,17,18,19,20,21,22]. Negro et al. found concentrations of total polyphenols higher than those detected in our samples; however, comparing the single compounds, the differences are mainly due to the very high levels of 1,5-di-O-caffeoylquinic acid in the authors’ samples, whereas the proportions of flavonoids and other caffeoylquinic acids were similar [19]. It should be noted that the authors reported 1,5-di-O-caffeoylquinic acid as cynarine; however, most of the paper identified cynarine as 1,3-di-O-caffeoylquinic acid [52]. In our work, we followed this classification, which was confirmed also by the analytical standards. According to the literature, flavonoids were most abundant in the leaves and caffeoylquinic acid was most abundant in the floral stems and heads, whereas outer bracts showed similar values for the two families [16,19,20,21,22,30]. Among the single biomolecules, chlorogenic acid was the most concentrated in stems, bracts, and heads, followed by 3,4 di-O-caffeoylquinic acid. In contrast, luteolin 7-O-rutinoside and 7-O-glucoside showed the highest levels in the leaves. Cynarine was found at low concentrations in all areas of the fresh artichoke. Gouveia et al. (2012) reported that cynarine is not naturally produced in artichoke, but that it originates during the preparation of juices and/or hydroalcoholic extracts [53]. Our data and those reported from other authors on the polyphenolic fraction extracted in water showed the presence of cynarine in significant concentrations in the natural matrix.

Different authors reported a significant variability in the concentration of biomolecules depending on the artichoke variety, season and harvesting time and site [16,18,19,20,21,22,23,24,25,26,27]. The biosynthesis of plant metabolites (flavonoids and hydroxycinnamates) is strongly influenced by climate changes and exposure to UV radiation; low temperatures and solar radiation levels can increase the development of electrons from photosynthesis, leading to the final number of polyphenols and ROS activity [52,54]. This fact can explain the high levels of flavonoids in the leaves and stems, being exposed for more time to the sun’s rays than the heads and its bracts (Table 3) [52].

The processing of artichoke in fresh-cut products generates a high amount of waste, considering both the plants remaining in the field and the bracts, stems and leaves removed during processing. Francavilla et al. reported that, after head harvesting, the residual artichoke biomass (Madrigal variety) left in the field accounted for about 33 tons per hectare (DW) [14]. Pesce et al. found artichoke by-products from Spinoso Sardo, Violetto di Sicilia, and Apollo varieties ranging from 5.1 tons per hectare (DW) to 13.8 tons per hectare (DW) [55]. The estimated amount of biomass waste in our study was 800 tons per year (DW). Waste elimination represents an important cost for the factories; therefore, its characterization and the development of green extraction methodologies is mandatory to obtain biomolecules that can be used as integrators for human and animals or pharmaceutical products.

Several authors reported the more effective extraction of polyphenols from food matrices using mixtures of water and ethanol using conventional and non-conventional techniques [6,15,22]. However, Santiago et al. (2020) found that the traditional extraction of polyphenols by utilizing a water–ethanol mixture (70:30 *v*/*v*) could lead to severe environmental impacts due to its secondary production processes [56]. The extraction with hot water proposed in this paper (80 °C, matrix-to-solvent ratio of 1:30, 120 min) represented a good compromise.

Dried waste and sun-dried plants showed a lower total polyphenol concentration. However, when considering the yield of dried samples compared to the fresh samples, the values of polyphenols recovered, and were on average 5.7 ± 17.2 times higher than the fresh biomass.

Drosou et al. reported an important reduction in the number of by-products but also an increase in the effectiveness of the recovery of bioactive compounds in the drying process for red grape pomace; however, they performed three different water extraction techniques (Soxhlet, microwave, and ultrasonic) on fresh and dried grape pomace, obtaining values for the dried samples that were 1.7, 2.12, and 2.58 times higher than those for the fresh samples [57]. Indeed, applying the right kinetic energy to a dried sample increases the hydration and swelling of the biomass, enlarging the pores of the cell wall, and enhancing the ability of the solvent to extract bioactive [58].

Changes in particle size can highly influence the extraction of active biomolecules from plant samples. Reducing the matrix size can lead to a high surface area, thus improving the interaction extraction solvent/matrix and the final mass transfer [59].

However, polyphenols are a well-known biomolecule involved in browning reactions in artichoke due to oxidative phenomena related to mechanical and thermal stress [60]. D_W_ showed a reduced quantitative polyphenolic concentration with respect to WD_W_ (Table 4). An oxidation process due to shredding probably occurred, neutralizing the contribution given by the increase in the particle surface of the matrix.

The yield obtained from WD_P_ showed a polyphenolic concentration 1.5 higher than the fresh plant. The qualitative profile of flavonoids was less affected (Table 4). The roots were the part of the plant with the highest concentration of cynarine (345 mg/kg) (Table 4).

Mena Garcia et al., after extraction in the water of outer bracts (Blanca de Tudela variety) using the microwave-assisted traction technique, found a concentration of total polyphenols accounting for about 2000 mg/kg FW, lower than our results [61]. Lavecchia et al., after the extraction of stems and outer bracts with different water/ethanol solutions, found a polyphenol content of 6300 mg/kg (FW) and 3000 mg/kg (FW), respectively [10]. However, most works do not consider the absolute waste produced from industrial processes. Only Ruiz Cano et al. evaluated the total polyphenol content (matrix to water ratio 1:200) in other artichoke waste sampled from other industrial processing steps. The samples differed in their thermal treatment (blanching and boiling), the bract position, and the cutting size. Bracts closer to the artichoke heart and boiled had the best-performing fraction [39].

As reported by other authors, the concentration of inulin in artichoke waste is considerable, with the roots showing the highest values [14,32,34,35]. However, for polyphenols, the extraction techniques, climatic conditions, and genetics strongly influence the production of inulin by plant matrices; moreover, after the maceration of artichoke roots belonging to different varieties (2 h, 80 °C, 1:6), the authors reported inulin values ranging from 0.8 g/100 g FW to 1.9 g/100 g FW [32]. Francavilla et al., after the extraction of roots (Madrigal variety) both by a conventional method (1 h, 85 °C, 1:10 matrix to solvent ratio) and Microwave-Assisted Extraction, found a concentration of inulin accounting for 6.1 g/100 g FW, slightly higher than that found in this work (Figure 2) [14].

Our data were higher than data found by Cuenca et al. in a blend of artichoke outer bracts and stems (0.58 g/100 g FW), and were consistent with values reported by Magda et al. after the maceration of the outer and inner bracts of the French Hyrious variety (2.18 g/100 g FW) [34,43].

The amount of fresh artichoke and waste considered in this study was comparable to a semi-industrial trial; to evaluate the scalability of an industrial process, it is essential to evaluate the average monthly waste load during the processing period. Based on the literature and real factory data, the average monthly waste generated for 100 tons, considering 20 working days, can be estimated at 5 tons/day. The critical point of the process is represented by the drying step, which, to avoid the loss of important biomolecules, should be performed at 30 °C. Artichoke is rich in fiber, which extends drying times. Therefore, only two complete processing steps/day (shredding, drying, extraction) should be carried out, and the plant should be set to process 2.5 tons/day/cycle.

## 5. Conclusions

Fresh samples and artichoke waste, fresh and dried and from the cultivar Tema, were subjected to green extractions, and the feasibility of recovering polyphenols and inulin was evaluated.

Artichoke generates a considerable amount of waste yearly in the field at the end of the season (5.2 tons per hectare DW) and in the industrial processing of fresh-cut products during the harvesting time (800 tons per year). Artichoke waste, fresh and dried, was investigated for its polyphenol and inulin content. The best operative conditions were achieved using the dried biomass extracted with water at 80 °C for 120 min and a matrix-to-solvent ratio of 1:30.

The data obtained when assessing different extraction conditions showed the presence of a considerable amount of high-value compounds, thus confirming that artichoke by-products may represent an important source of health-promoting compounds suitable for producing phyto-complexes for food, pharmaceutical and cosmeceutical purposes.

Both conventional and innovative extraction technology shows higher extraction rates after the size of the plant material is reduced to facilitate mass transfer. However, due to possible oxidative effects and to preserve the polyphenolic fraction, it is necessary to perform the chopping process in controlled techniques. To reduce labor and energy costs, the remaining plants in the field should be left to dry in the sun until the end of the crop cycle before being collected and processed together with the roots.

Thanks to the absence of organic solvents and energy and labor saving during the sample treatment, the work scheme proposed in this article could provide a scalable, practical and economic opportunity for companies and farmers involved in the cultivation and industrial processing of artichokes.

## Figures and Tables

**Figure 1 foods-14-00013-f001:**
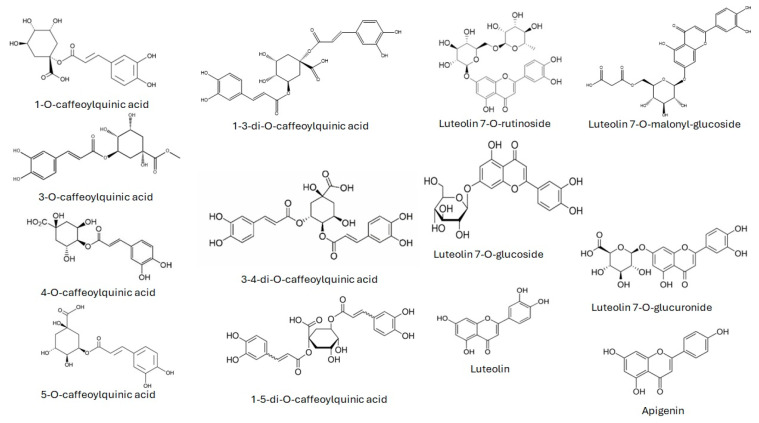
Caffeoylquinic acids, luteolin derivatives, and apigenin identified in the artichoke’s extracts.

**Figure 2 foods-14-00013-f002:**
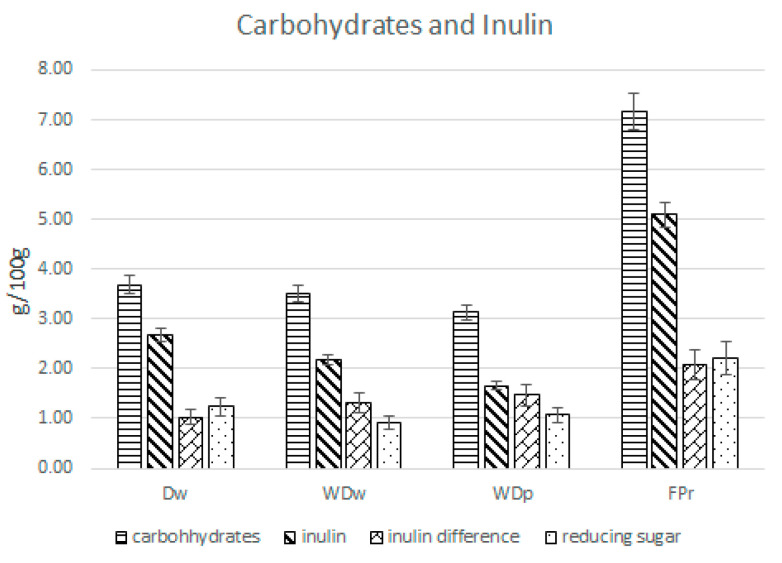
Carbohydrates, inulin and reducing sugar concentration (g/100 g FW) in artichoke by-products extracted at 80 °C for 120 min. WD_W_: whole dry waste; D_W:_ dry waste; WD_P:_ whole dry plant; FP_r_: fresh plant from the field (roots).

**Table 1 foods-14-00013-t001:** The moisture and ash content of fresh and dried artichoke samples, processed during the trials.

Sample	Moisture (% ± RSD%)	Ash (% ± RSD%, F_W_)
F_H_ *	83.0 ± 1.9	1.0 ± 9.8
F_B_	82.0 ± 2.6	1.4 ± 6.8
F_S_	85.0 ± 1.6	1.4 ± 10.6
F_L_	76.1 ± 0.3	1.8 ± 16.8
F_W_	84.8 ± 0.8	1.1 ± 19.3
FPs_L_	81.2 ± 2.2	1.2 ± 4.2
FP_R_	62.2 ± 1.2	1.6 ± 0.7
WD_W_	15.5 ± 4.2	1.2 ± 2.0
D_W_	12.4 ± 5.4	1.1 ± 1.3
WD_P_	16.5 ± 2.7	1.0 ± 4.3

* F_H_: heads; F_B_: outer bracts; F_S_: stems: F_L_: leaves; WD_W_: whole dry waste; F_W_: fresh waste; D_W_: dry waste; WD_P_: whole dry plant; FP_SL_: fresh plant from the field (stems + leaves); FP_R_: fresh plant from the field (roots).

**Table 2 foods-14-00013-t002:** HPLC-DAD polyphenol profile (mg/kg ± RSD%) of fresh bracts (F_B_) extracted in hot water (80 °C) with different matrix/solvent ratios and extraction times.

Matrix/Solvent	1:5	1:5	1:10	1:20	1:30
Extraction Time	40 min	120 min	120 min	120 min	120 min
	mg/kg ± RSD%				
1-O-caffeoylquinic acid	335.7 ^y^ ± 6.1 ^a^	323.5 ± 2.6 ^a^	375.4 ± 8.1 ^a^	417.7 ± 7.2 ^a^	360.0 ± 17.6 ^a^
3-O-caffeoylquinic acid	136.8 ± 9.8 ^a^	192.1 ± 7.1 ^a^	261.2 ± 6.0 ^ab^	336.0 ± 13.6 ^b^	408.5 ± 9.3 ^c^
4-O-caffeoylquinic acid	247.2 ± 2.8 ^a^	265.6 ± 12.0 ^a^	313.3 ± 6.8 ^ab^	381.0 ± 7.8 ^b^	277.3 ± 8.4 ^a^
5-O-caffeoylquinic acid (chlorogenic)	1390.4 ± 8.2 ^a^	1323.5 ± 7.7 ^a^	1287.2 ± 5.6 ^a^	1319.4 ± 13.6 ^a^	1901.5 ± 4.9 ^b^
1,3-di-O-caffeoylquinic acid (cynarin)	319.7 ± 5.9 ^a^	566.4 ± 6.5 ^b^	676.8 ± 3.2 ^b^	691.6 ± 11.8 ^b^	634.5 ± 4.8 ^b^
3,4 di-O-caffeoylquinic acid	1276.5 ± 7.8 ^a^	1413.7 ± 11.3 ^b^	1234.8 ± 0.6 ^a^	1399.1 ± 1.6 ^ab^	1536.8 ± 8.2 ^b^
1,5-di-O-caffeoylquinic acid	40.8 ± 9.9 ^a^	122.7 ± 11.3 ^b^	98.1 ± 7.7 ^b^	110.6 ± 2.4 ^b^	167.6 ± 0.9 ^c^
Other caffeoylquinic acid	154.1 ± 5.7 ^a^	300.4 ± 1.5 ^b^	281.4 ± 3.7 ^b^	296.6 ± 5.4 ^b^	538.3 ± 0.7 ^c^
**Total caffeoylquinic acids**	**3901.2 ± 3.6** ^a^	**4508.6 ± 4.2** ^ab^	**4528.1 ± 2.7** ^ab^	**4952.0 ± 5.3** ^b^	**5824.4 ± 2.5** ^c^
Luteolin 7-O-rutinoside	160.0 ± 15.1 ^a^	166.7 ± 6.0 ^a^	182.2 ± 3.3 ^a^	219.4 ± 10.3 ^b^	249.0 ± 4.4 ^b^
Luteolin 7-O-glucoside	75.6 ± 14.4 ^a^	83.4 ± 6.1 ^a^	88.5 ± 6.6 ^a^	158.1 ± 0.7 ^b^	163.2 ± 5.0 ^b^
Luteolin 7-O-glucuronide	99.1 ± 4.4 ^a^	110.5 ± 1.7 ^a^	118.3 ± 8.1 ^a^	117.5 ± 5.5 ^a^	125.6 ± 3.4 ^b^
Luteolin 7-O-malonylglucoside	150.8 ± 7.3 ^a^	157.4 ± 1.2 ^a^	135.7 ± 3.9 ^a^	156.9 ± 5.5 ^a^	215.7 ± 4.2 ^b^
Luteolin	88.6 ± 8.1 ^a^	97.5 ± 3.2 ^a^	101.4 ± 1.0 ^a^	101.8 ± 18.5 ^a^	120.0 ± 2.3 ^b^
Apigenin	68.3 ± 1.2 ^a^	67.3 ± 13.7 ^a^	60.6 ± 2.0 ^a^	67.8 ± 5.1 ^a^	69.2 ± 4.0 ^a^
**Total flavonoids**	**627.7 ± 6.3** ^a^	**682.8 ± 1.4** ^a^	**686.8 ± 1.0** ^a^	**821.6 ± 1.4** ^b^	**943.2 ± 1.7** ^c^
**Total polyphenols**	**4528.9 ± 4.0** ^a^	**5191.4 ± 3.5** ^a^	**5381.8 ± 3.8** ^a^	**5773.5 ± 4.4** ^b^	**6767.6 ± 2.3** ^c^

^y^ Different letters between columns indicate statistical significance among rows at *p* < 0.01.

**Table 3 foods-14-00013-t003:** Polyphenol profile of the different portions of fresh artichoke extracted with a solvent-to-matrix ratio of 1:30 for 2 h in hot water (80 °C) and in MEOH/H_2_O 80:20 for 2 h at room temperature. ND: not detected.

	F_B_ *		F_S_		F_H_		F_L_	
	mg/kg ± RSD%							
Extraction	Hot Water ^y^	MeOH/H_2_O	Hot Water	MeOH/H_2_O	Hot Water	MeOH/H_2_O	Hot Water	MeOH/H_2_O
1-O-caffeoylquinic acid	360.0 ± 17.6 ^a(a)^	102.3 ± 16.4 ^b(x)^	280.4 ± 9.4 ^a(a)^	105.4 ± 9.3 ^b(x)^	728.1 ± 3.6 ^a(b)^	198.7 ± 10.0 ^b(y)^	327.3 ± 15.2 ^a(a)^	100.4 ± 15.4 ^b(x)^
3-O-caffeoylquinic acid	408.5 ± 9.3 ^a(a)^	61.5 ± 5.8 ^b(x)^	266.5 ± 3.8 ^a(b)^	87.3 ± 8.3 ^b(x)^	248.5 ± 13.5 ^a(b)^	64.0 ± 4.7 ^b(x)^	263.8 ± 16.2 ^a(b)^	62.7 ± 6.7 ^b(x)^
4-O-caffeoylquinic acid *	277.3 ± 8.4 ^(a)^	ND	482.1 ± 18.5 ^(b)^	ND	577.0 ± 15.5 ^(b)^	ND	624.1 ± 17.8 ^(c)^	ND
5-O-caffeoylquinic acid (chlorogenic)	1901.5 ± 4.9 ^a(a)^	755.8 ± 17.0 ^b(x)^	6879.7 ± 4.6 ^a(b)^	3441.5 ± 19.3 ^b(y)^	5492.5 ± 3.9 ^a(b)^	2457.0 ± 12.6 ^b(y)^	6908.8 ± 13.2 ^a(b)^	1701.8 ± 9.5 ^b(z)^
1,3-di-O-caffeoylquinic acid (cynarin)	634.5 ± 4.8 ^a(a)^	61.6 ± 0.3 ^b(x)^	254.1 ± 19.3 ^a(b)^	72.5 ± 4.0 ^b(x)^	514.7 ± 16.3 ^a(a)^	73.7 ± 4.1 ^b(x)^	148.7 ± 6.7 ^a(b)^	80.0 ± 2.8 ^b(y)^
3,4 di-O-caffeoylquinic acid	1536.8 ± 8.2 ^a(a)^	583.6 ± 17.4 ^b(x)^	972.5 ± 21.3 ^a(b)^	275.8 ± 13.8 ^b(y)^	1578.4 ± 14.5 ^a(a)^	1301.9 ± 6.0 ^b(z)^	1503.5 ± 10.5 ^a(a)^	348.9 ± 14.5 ^b(xy)^
1,5-di-O-caffeoylquinic acid	167.6 ± 0.9 ^a(a)^	100.4 ± 5.7 ^b(x)^	507.8 ± 10.0 ^a(b)^	181.2 ± 12.1 ^b(x)^	592.4 ± 10.5 ^a(b)^	163.4 ± 14.1 ^b(x)^	254.3 ± 16.7 ^a(c)^	168.2 ± 8.1 ^b(x)^
Other di-caffeoylquinic acids	538.3 ± 0.7 ^(a)^	ND	2091.9 ± 15.6 ^(b)^	ND	1812.4 ± 13.7 ^(b)^	ND	104.2 ± 10.5 ^(c)^	ND
**Total caffeoylquinic acids**	**5824.4 ± 2.5** ^a(a)^	**1665.2 ± 14.9** ^b(x)^	**11,735.0 ± 8.0** ^a(b)^	**4163.6 ± 17.1** ^b(y)^	**11,544.0 ± 5.1** ^a(b)^	**4258.7 ± 9.3** ^b(y)^	**11,070.6 ± 12.9** ^a(b)^	**2462.0 ± 7.0** ^b(x)^
Luteolin 7-O-rutinoside	249.0 ± 4.4 ^a(a)^	14.3 ± 9.8 ^b(x)^	735.8 ± 19.4 ^a(b)^	94.3 ± 8.8 ^b(y)^	541.2 ± 10.6 ^a(c)^	12.8 ± 6.3 ^b(x)^	715.4 ± 13.1 ^a(b)^	216.7 ± 17.0 ^b(z)^
Luteolin 7-O-glucoside	163.2 ± 5.0 ^a(a)^	30.5 ± 15.9 ^b(x)^	151.3 ± 16.4 ^a(a)^	123.0 ± 2.0 ^b(y)^	240.6 ± 16.0 ^a(b)^	44.2 ± 12.5 ^b(x)^	455.8 ± 13.8 ^a(c)^	123.8 ± 12.1 ^b(y)^
Luteolin 7-O-glucuronide	125.6 ± 3.4 ^a(a)^	38.1 ± 2.3 ^b(x)^	34.3 ± 9.2 ^(b)^	ND	53.9 ± 9.3 ^a(b)^	14.9 ± 17.0 ^b(y)^	53.4 ± 16.9 ^a(b)^	ND
Luteolin 7-O-malonylglucoside	215.7 ± 4.2 ^a(a)^	91.0 ± 5.3 ^b(x)^	211.6 ± 14.8 ^a(a)^	142.2 ± 6.2 ^b(y)^	134.4 ± 13.0 ^a(b)^	72.2 ± 14.8 ^b(x)^	139.9 ± 13.5 ^a(b)^	111.6 ± 2.0 ^a(x)^
Luteolin	120.0 ± 2.3 ^a(a)^	5.2 ± 9.5 ^b(x)^	165.1 ± 16.6 ^(b)^	ND	115.3 ± 4.3 ^(a)^	<LOQ	127.9 ± 14.2 ^a(a)^	78.0 ± 2.6 ^b(x)^
Apigenin	69.2 ± 4.0 ^(a)^	<LOQ	67.4 ± 8.4 ^(a)^	<LOQ	69.9 ± 12.3 ^(a)^	<LOQ	56.6 ± 13.3 ^(a)^	<LOQ
**Total flavonoids**	**943.2 ± 1.7** ^a(a)^	**356.0 ± 2.1** ^b(x)^	**1516.7 ± 13.2** ^a(a)^	**813.3 ± 8.4** ^b(y)^	**1155.4 ± 4.4** ^a(a)^	**378.4 ± 15.2** ^b(x)^	**1549.0 ± 9.6** ^a(a)^	**649.8 ± 11.5** ^b(y)^
**Total polyphenols**	**6767.6 ± 2.3** ^a(a)^	**2021.3 ± 12.6** ^b(x)^	**13,100.4 ± 8.2** ^a(b)^	**4976.8 ± 15.4** ^b(y)^	**12,699.3 ± 5.0** ^a(b)^	**4637.2 ± 9.6** ^b(y)^	**12,619.6 ± 12.4** ^a(b)^	**3111.8 ± 8.0** ^b(xy)^

* F_H_: heads; F_B_: outer bracts; F_S_: stems: F_L_: leaves. ^y^ Different letters between columns indicate statistical significance at *p* < 0.01. Letters without parentheses relate to comparisons of the effects of the extraction solvent, within each artichoke part. Letters in parentheses relate to comparisons of the influence of the artichoke fraction among each extraction solvent.

**Table 4 foods-14-00013-t004:** The polyphenol profile (g/100 g FW) of artichoke waste from industrial processing and collected in the field. WD_W_: whole dry waste; F_W_: fresh waste; D_W_: dry waste; WD_P_: whole dry plant; FP_SL_: fresh plant from the field (stems + leaves); FP_R_: fresh plant from the field (roots).

Sample	WD_W_	F_W_	D_W_	FP_SL_	FP_R_	WD_P_
	mg/kg ± RSD%					
1-O-caffeoylquinic acid	144.9 ± 6.69	423.9 ± 11.6	41.0 ± 16.3	218.7 ± 7.8	111.6 ± 14.8	96.0 ± 2.6
3-O-caffeoylquinic acid	250.0 ± 16.6	296.8 ± 13.5	79.7 ± 9.4	347.8 ± 13.2	372.8 ± 3.2	269.9 ± 3.1
4-O-caffeoylquinic acid	141.4 ± 9.17	490.1 ± 13.0	89.9 ± 4.9	575.6 ± 5.8	274 ± 9.1	143.9 ± 9.4
5-O-caffeoylquinic acid (chlorogenic)	1574.5 ± 2.43	5295.6 ± 9.4	246.6 ± 18.1	6318.0 ± 4.2	747.6 ± 8.1	1047.7 ± 6.0
1,3-di-O-caffeoylquinic acid (cynarin)	229.2 ± 19.2	388.0 ± 11.1	132.7 ± 14.7	227.5 ± 17.0	349.6 ± 2.2	213.0 ± 2.5
3,4-di-O-caffeoylquinic acid	148.2 ± 1.42	1397.8 ± 15.6	59.8 ± 14.9	966.6 ± 14.9	131.6 ± 8.2	150.5 ± 4.4
1,5-di-O-caffeoylquinic acid	804.6 ± 13.4	380.5 ± 6.3	488.0 ± 9.2	395.8 ± 4.9	175.2 ± 13.9	528.4 ± 11.3
Other caffeoylquinic acid	797.5 ± 0.9	1370.7 ± 18.4	777.2 ± 3.0	1709.3 ± 15.4	267.4 ± 7.9	640.0 ± 17.3
**Total caffeoylquinic acids**	**4090.2 ± 1.1**	**10,043.5 ± 6.6**	**1914.9 ± 2.0**	**10,759.3 ± 18.1**	**2429.8 ± 10.1**	**3089.4 ± 15.9**
Luteolin 7-O-rutinoside	121.9 ± 2.1	560.5 ± 6.6	68.3 ± 8.8	630.5 ± 5.5	255.0 ± 4.8	140.9 ± 3.5
Luteolin 7-O-glucoside	96.7 ± 15.9	252.7 ± 3.7	56.2 ± 7.4	356.5 ± 3.7	315.8 ± 9.0	126.0 ± 10.2
Luteolin 7-O-glucuronide	20.5 ± 6.0	66.8 ± 8.8	20.4 ± 13.7	81.8 ± 8.6	81.5 ± 7.3	46.0 ± 7.2
Luteolin 7-O-malonylglucoside	125.2 ± 8.5	175.4 ± 18.7	54.0 ± 9.0	160.4 ± 7.1	250.6 ± 9.8	154.6 ± 8.1
Luteolin	63.8 ± 2.8	132.1 ± 17.1	38.1 ± 12.3	162.6 ± 10.1	66.9 ± 5.4	102.7 ± 11.5
Apigenin	17.1 ± 8.3	65.9 ± 9.6	9.8 ± 15.4	79.8 ± 1.5	15.0 ± 4.3	27.0 ± 6.8
**Total flavonoids**	**445.3 ± 4.8**	**1253.4 ± 4.2**	**246.7 ± 3.5**	**1471.6 ± 5.5**	**984.8 ± 10.1**	**597.1 ± 6.4**
**Total polyphenols**	**4535.5 ± 0.5**	**11,296.9 ± 5.7**	**2161.6 ± 2.1**	**12,230.9 ± 13.2**	**3414.6 ± 11.4**	**3686.5 ± 9.1**

## Data Availability

The original contributions presented in the study are included in the article, further inquiries can be directed to the corresponding author.

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
