# Peer review of "Extraction and Characterization of Artichoke (Cynara cardunculus L.) Solid Waste from the Industrial Processing of Fresh-Cut Products for Nutraceutical Use"

_foods, 2024, doi:10.3390/foods14010013_

Round 1
Reviewer 1 Report
Comments and Suggestions for Authors
After reviewing the manuscript, I have the following comments and suggestions:
1. Many studies cited in the manuscript have explored polyphenol and inulin content in artichoke waste using similar methodologies. The paper does not sufficiently highlight how its findings advance the field beyond what is already known.
2. Methodology: Correct: Plant material
3. Methodology: Please provide additional details about the extraction vessel used.
4. Methodology (item 2.6): The authors describe inulin calculated by subtracting reducing sugars from total carbohydrates. This indirect approach can introduce inaccuracies if other carbohydrates or interferents are present. A more precise analytical technique, such as HPLC or TLC, could have been used.
5. Further clarity on the reproducibility of the optimized extraction method could enhance the robustness of the study. For example, consider including a detailed statistical analysis comparing green and conventional extraction techniques across various artichoke parts.
6. The authors should discuss the scalability of the proposed process.
7. The conclusion is overly lengthy and should be rewritten to provide a concise and clear summary of the key findings and their implications. Focus on the main results, significance, and potential applications while omitting unnecessary details or repetitive information.
Author Response
1) Many studies cited in the manuscript have explored polyphenol and inulin content in artichoke waste using similar methodologies. The paper does not sufficiently highlight how its findings advance the field beyond what is already known.
Answer: Some authors reported the influence of the industrial processes on waste not considering the products deriving from the field, others studied artichokes taken in the field and / or bought on the market not considering the industrial waste.
This article, for the first time, evaluated the whole artichoke supply chain, including the fresh artichokes from the fields, the waste from industrial processing and the plant left in the field at the end of the harvest season. This statement was added at the end of the introduction (Line: 72-74).
2) Methodology: Correct: Plant material
Answer: The title was corrected accordingly.
3) Methodology: Please provide additional details about the extraction vessel used.
Answer: The information was implemented accordingly (Line 127).
4) Methodology (item 2.6): The authors describe inulin calculated by subtracting reducing sugars from total carbohydrates. This indirect approach can introduce inaccuracies if other carbohydrates or interferents are present. A more precise analytical technique, such as HPLC or TLC, could have been used.
Answer: We agree with the reviewer, however, this procedure was used to confirm data obtained by inulin extraction, according to several methods reported in literature data.
5) Further clarity on the reproducibility of the optimized extraction method could enhance the robustness of the study. For example, consider including a detailed statistical analysis comparing green and conventional extraction techniques across various artichoke parts.
Answer: Statistical information was added in Table 3, accordingly.
6) The authors should discuss the scalability of the proposed process.
Answer: the discussion was improved, accordingly.
7) The conclusion is overly lengthy and should be rewritten to provide a concise and clear summary of the key findings and their implications. Focus on the main results, significance, and potential applications while omitting unnecessary details or repetitive information.
Answer: the conclusion was modified, accordingly.
Reviewer 2 Report
Comments and Suggestions for Authors
In this manuscript, the authors aimed to characterize the content of bioproducts (polyphenolic compounds and inulin) in the different parts of the globe artichoke (outer and inner bracts, stem, choke, heart and roots) as the fresh cut products commercially available and from the waste collected during the processing step and left in the field after harvesting. The challenge will provide useful information for further industrial use of the unused part of this plant in the future. Most of the data were properly collected and the results were well discussed. Please address the minor issues raised by this reviewer.
Introduction section
1. P2, line 77. Please define the rationale for the importance of green extraction in the Introduction section. This reviewer was able to understand this after reading lines 311-329 in the Discussion section.
Materials and Methods section
2. P3, lines 136-137. Typo? A word “derivates” should be “derivatives”.
3. P4, lines 149-153. Please describe in detail how the inulin and carbohydrate contents were determined.
Results section
4. Tables 1-4 and Figure 2. Please define the abbreviation, e.g. FH, FB, FS…, again at each footnote (see lines 100-112, again).
5. P4, line 174. Typo? The word “lutein” should be “luteolin”.
6. Figure 1. Typo? The words “luteolina” and “apigenina” should be “luteolin” and “apigenin”, respectively. Also, “malonil” should be “malonyl”.
7. Page 5, lines 183-198. Please check that the ratio (%) values in the text are correct from those in the table. For example, “6768.2±2.4 (line 189)” in the text looks different from “6767.6±2.3” in Table 2. Also, please check not only Table 2 but also the others to see if they in the text agree with the original data appearing in the tables (see also 171.1% and 314.2% in line 205).
8. Table 3. Typo? A word “MEOH” should be “MeOH”.
9. Page 7, lines 224-228. The authors have mentioned about Table 4 in this paragraph. Please insert (Table 4) on line 226 and delete it that appeared on line 233.
10. Page 8, Figure 2. Please show the unit of the y-axis.
11. Page 10. Line 349. Typo? The word “blenching” should be “blanching”.
12. Page 10. Line 377. Typo? Please add a period.
Author Response
In this manuscript, the authors aimed to characterize the content of bioproducts (polyphenolic compounds and inulin) in the different parts of the globe artichoke (outer and inner bracts, stem, choke, heart and roots) as the fresh cut products commercially available and from the waste collected during the processing step and left in the field after harvesting. The challenge will provide useful information for further industrial use of the unused part of this plant in the future. Most of the data were properly collected and the results were well discussed. Please address the minor issues raised by this reviewer.
Introduction section
1) P2, line 77. Please define the rationale for the importance of green extraction in the Introduction section. This reviewer was able to understand this after reading lines 311-329 in the Discussion section.
Answer: the introduction was improved, accordingly.
Materials and Methods section
2) P3, lines 136-137. Typo? A word “derivates” should be “derivatives”.
Answer: the word was corrected, accordingly.
3) P4, lines 149-153. Please describe in detail how the inulin and carbohydrate contents were determined.
Answer: the detailed description was added, accordingly.
Results section
4) Tables 1-4 and Figure 2. Please define the abbreviation, e.g. FH, FB, FS…, again at each footnote (see lines 100-112, again).
Answer: the abbreviations were defined and added in Tables 1-4 and Figure 2, accordingly.
5) P4, line 174. Typo? The word “lutein” should be “luteolin”.
Answer: the word lutein was corrected, accordingly.
6) Figure 1. Typo? The words “luteolina” and “apigenina” should be “luteolin” and “apigenin”, respectively. Also, “malonil” should be “malonyl”.
Answer: Figure 1 was corrected, accordingly.
7) Page 5, lines 183-198. Please check that the ratio (%) values in the text are correct from those in the table. For example, “6768.2±2.4 (line 189)” in the text looks different from “6767.6±2.3” in Table 2. Also, please check not only Table 2 but also the others to see if they agree with the original data appearing in the tables (see also 171.1% and 314.2% in line 205).
Answer: the text was checked and corrected, accordingly.
8) Table 3. Typo? A word “MEOH” should be “MeOH”.
Answer: Table 3 was corrected, accordingly.
9) Page 7, lines 224-228. The authors have mentioned about Table 4 in this paragraph. Please insert (Table 4) on line 226 and delete it that appeared on line 233.
Answer: The text was corrected, accordingly.
10) Page 8, Figure 2. Please show the unit of the y-axis.
Answer: The figure was implemented, accordingly.
11) Page 10. Line 349. Typo? The word “blenching” should be “blanching”.
Answer: The text was corrected, accordingly.
12) Page 10. Line 377. Typo? Please add a period.
Answer: The text was corrected, accordingly.
Round 2
Reviewer 1 Report
Comments and Suggestions for Authors
The authors did the requested changes and the manuscript can be published by Foods